# Ameliorating impact of coenzyme Q10 on the profile of adipokines, cardiomyopathy, and hematological markers correlated with the glucotoxicity sequelae in diabetic rats

**Yousif Jameel Jbrael**[1], **Badraldin Kareem Hamad**[1,2] *

1 Department of Pharmacology and Toxicology, College of Pharmacy, Hawler Medical University, Erbil, Iraq,
2 University of Kurdistan Hawler (UKH), School of Medicine, Erbil, Iraq

* Badraldin.hamad@hmu.edu.krd

## Abstract

### Background

In diabetes, high blood glucose induces glucotoxicity, resulting in the further damage of pancreatic beta-cells and then precipitating diabetic complications. This study was aimed to investigate the relationship between glucotoxicity with the level of adipokines, diabetic cardiomyopathy, and hematological markers. Moreover, the study examined the potential modulatory effect of coenzyme Q10 (CoQ10) on the aforementioned markers associated with the sequelae of diabetes mellitus.

### Material and methods

Twenty-four male rats were randomly assigned to receive an injection of STZ to induce diabetes (n = 16) or to remain uninduced (n = 8). The hyperglycemic status was induced in fasting rats by single intraperitoneal injection of STZ (45 mg /kg b.w.) dissolved in citrate buffer (pH 4.5). Three days after STZ injection, rats were divided into three groups; Normal control group (A), Diabetic control group (B), and CoQ10- treated diabetic group (C). The group (C) was fed with the basal diet supplemented with 5 g of CoQ10 per kilogram of diet for three weeks after the diabetes induction. After 21 days, the blood and serum samples were taken to conduct biochemical analyses. Blood glucose was determined by Blood Glucose Monitoring System. Adipokines or cytokines were evaluated by ELISA from a serum sample. Cardiac myopathy biomarkers were estimated by UP-Converting Phosphor Immunoassay Analyzer, and hematological parameters were measured by automatic hematology analyzer.

### Results

In hyperglycemic rats, the level of fasting blood glucose, and serum level of resistin, omentin, TNF-α, and cardiomyopathy biomarkers significantly increased (P < 0.05). The treatment with CoQ10 significantly decreased the profile of adipokines and cardiomyopathy markers (cardiac enzymes and LPPLA2) in diabetic rats and also reduced glucose levels (P

**Data Availability Statement:** All relevant data are within the manuscript.

**Funding:** The authors received no specific funding for this work.

**Competing interests:** The authors have declared that no competing interests exist.

< 0.05). Lymphocyte percentages significantly decreased while significant increases were observed in granulocytes and MID percentages in hyperglycemic rats.

## Conclusion

Diabetic rats had higher serum levels of adipokines and cardiomyopathy markers. Among the hematological markers, GRA% and MID% increased while LYM% decreased. The profile of adipokines and cardiomyopathy markers improved when CoQ10 was supplemented. The study suggests that CoQ10 may have a beneficial effect on improving diabetic complications.

## 1. Introduction and aims of the study

Diabetes mellitus (DM), characterized by elevated blood sugar levels, encompasses a range of heterogeneous metabolic disorders. It stands as a leading cause of both mortality and morbidity worldwide, contributing significantly to healthcare expenditures [1, 2]. In general, the adverse effects of hyperglycemia are divided into macrovascular complications (coronary artery disease, stroke, and peripheral arterial disease) and microvascular complications (diabetic nephropathy, retinopathy, and neuropathy) [3]. Macrovascular issues are mostly caused by the atherosclerotic constriction of large arteries and veins, which results in cardiovascular, cerebrovascular, and peripheral arterial disorders [4]. In the majority of the diabetic population, cardiovascular disease is the predominant cause of death, approximately 80% of diabetics aged 65 and older die from coronary heart disease [5]. Cerebrovascular disorders, such as ischemia and stroke, affect 20–40% of diabetic persons due to atherosclerotic narrowing of the intracranial arteries [6]. Peripheral artery disease (PAD) is the atherosclerotic occlusive disease of the lower limbs, which is frequently accompanied with a high risk of amputation of the involved extremities. One of the independent risk factors for the development of PAD is diabetes mellitus. Particularly, diabetic patients frequently develop critical limb ischemia, the advanced phase of PAD characterized with rest pain and leading to permanent functional impairment [7, 8]. In 50% of cases, PAD leads to diabetic foot ulcers, and these percentages may increase in the future [9]. While macrovascular abnormalities are commonly observed in individuals with diabetes, the root cause of these conditions often lies in the progression of microvascular complications. Notably, research has highlighted that the occurrence of microvascular problems in the heart and brain coincides with the development of macrovascular diseases in patients with diabetes [10]. Microangiopathy, retinopathy, nephropathy, and neuropathy are all microvascular complications caused by hyperglycemia in small blood vessels, such as capillaries. Increased vascular permeability, neovascularization in the retina, and retinal thickness are the hallmarks of diabetic retinopathy, which causes vision loss. In individuals aged 20–74, diabetic retinopathy is the leading cause of new occurrences of blindness. Within the first two decades of diabetes, nearly all patients with type 1 diabetes and more than sixty percent of patients with type 2 diabetes develop retinopathy [11, 12]. Diabetic nephropathy is a significant healthcare concern that occurs up to fifty percent of diabetic persons. It is a main reason of end-stage kidney disease, which necessitates treatment with dialysis or transplantation, and is correlated with substantially increased cardiovascular morbidity and death [13]. Neuropathy is the most widespread complication of prediabetes and diabetes, with distal symmetric polyneuropathy being the most prevalent. Over time, at least fifty percent of people with diabetes experience diabetic neuropathy. Distal polyneuropathy diminishes the sensitivity of the extremities of the limbs, such as hands, legs, and feet, resulting in diabetic ulcers. Due to

loss of sensation, diabetic patients are sometimes unaware to injuries to their limbs, preventing them from receiving treatment. This circumstance may result in the development of gangrene, and eventually necessitating amputation of the involved limbs [14, 15].

While numerous investigations have been done to clarify the molecular pathways hidden behind the development of diabetes-related complications, their exact pathophysiological mechanisms remain unknown. Despite the fact that numerous studies have attempted to clarify the molecular pathways behind the development of diabetes complications, their precise pathophysiology remains poorly understood [16–18]. There is a general consensus that oxidative stress is one of the serious contributors to the progression of diabetes complications [19]. Oxidative stress is a critical aspect for developing cardiovascular and microvascular complications of diabetes. Hyperglycemia induced oxidative stress in the pattern of an increase in reactive oxygen/nitrogen species (ROS/RNS) and consumption of antioxidant defenses, such as catalase, superoxide dismutase, and glutathione peroxidase, has a pivotal function in the pathogenesis of DM and its complications [20–22]. Due to hyperglycemia-induced activation of the polyol pathway and production of AGEs and ROS, antioxidants play a role in the treatment of diabetes. One of them is alpha-lipoic acid, which is a powerful antioxidant in the prevention of diabetes and its consequences. It is capable of regenerating antioxidants such as glutathione, vitamin E, Coenzyme Q10, and vitamin C [23]. Curcumin is another antioxidant that could be utilized to alleviate indirect diabetes complications. In situations of hyperglycemia, it plays an active participation in the regulation of increased oxidative stress, protein glycation, and glucose metabolism [24, 25].

Ubiquinone (Coenzyme Q10 or CoQ10), known as an intracellular antioxidant, is an essential and famous cofactor for ATP production in the mitochondrial electron transport chain [26]. CoQ10 supplementation appears to improve mitochondrial function and provide antioxidant defense for tissues and organs impacted by a variety of pathophysiological disorders [27].

Hence, further analysis may be required to clarify the influence of CoQ10 supplementation on the levels of some new adipokines, inflammatory and cardiomyopathy markers associated with acute hyperglycemia in STZ-induced diabetic rats.

In diabetic rats, studies have identified increased consumption of the antioxidant defense system and heightened lipid peroxidation. Recent research has emphasized the pivotal role of oxidative stress in the development of numerous diabetic complications, highlighting its central and critical involvement in their etiology [20, 21]. Hyperglycemia raises the generation of advanced glycation end products (AGEs) by intensifying nonenzymatic glycation and attaching glycation end products to their receptor, resulting in excessive formation of intracellular reactive oxygen species (ROS) by nicotinamide adenine dinucleotide phosphate oxidase (NADPH oxidase). Similarly, protein kinase C is activated by diacylglycerol formation (which its level raised in diabetes), resulting in the ensuing formation of ROS through NADPH oxidase [28, 29]. When ROS levels rise, they begin to have detrimental effects on critical cellular structures, including nucleic acids, proteins, and lipids [30]. They have the ability to oxidize cell structures (nucleic acids, proteins, and lipids) and generate toxic byproducts leading to tissue dysfunction. Additionally, they modify the structures of biological molecules and even break them [31]. Several investigations have revealed that antioxidant treatment ameliorates diabetic complications [32]. Recently, researchers have been intrigued by examining and investigating the extraction of natural antioxidants to substitute artificial antioxidants [33, 34]. Consequently, additional research is necessary to determine the natural source of antioxidants in order to improve public health among patients with diabetes.

It was claimed that growing data supports the defensive effect of dietary antioxidants as a promising adjuvant treatment for postponing or preventing diabetic complications [35]. CoQ10 is a highly lipophilic antioxidant that is also involved in the recycling and regenerating

of other antioxidants, including β-carotene, tocopherol, and ascorbate [36, 37]. CoQ10, as the third-most popular nutritional supplement and a possible candidate to treat a variety of non-communicable diseases that are among the top 10 causes of mortality worldwide, has drawn increased attention over the years [38].

Several clinical investigations have demonstrated that oral dosing of CoQ10 has beneficial impacts on a variety of conditions linked with reduced CoQ10 levels and elevated oxidative stress, including mitochondrial, neurodegenerative, and cardiovascular diseases [39–42].

CoQ10 may be beneficial in reversing endothelial dysfunction and protecting against diabetic vasculopathy [43]. Appropriately, the pharmacologists emphasize the potential benefit of COQ10 as an adjunct to conventional anti-diabetic medications.

No study or very little studies are available to evaluate the correlation of glucotoxicity with our study biomarkers (some endogenous ligands, particularly adipokine biomarkers, vascular inflammation marker (Lipoprotein-associated phospholipase A2 [LPPLA2]), hematological variables, and cardiomyopathy biomarkers). There is no research linking CoQ10's antioxidant activity to endogenous ligands or biomarkers which having role in the mechanism of diabetes complications.

The present study examines the association between the state of glucotoxicity and the levels of specific endogenous pharmacologically active ligands, particularly adipokine biomarkers (for example, omentin, resistin, and TNF-α), vascular inflammation markers (such as Lipoprotein-associated phospholipase A2 (LPPLA2)), and hematological parameters. A further goal of the study is to evaluate the effect of glucotoxicity on the heart by measuring serum levels of diabetic cardiomyopathy biomarkers such as cardiac troponin I (cTnL) and Creatine kinase-MB (CK-MB). In addition, we investigated how CoQ10 affects levels of markers mentioned earlier. We propose that there is a correlation between glucotoxicity and levels of our examined ligand and biomarkers. Also, we hypothesize that there is a correlation between antioxidant effect of CoQ10 with the levels of above-mentioned ligands or biomarkers in favor of attenuating diabetic complications.

It is important to investigate the influence of high blood sugar on biochemical parameters as well as the impact of medications on their levels. Despite the fact that various studies have produced contradictory results regarding the beneficial effect of CoQ10, those that support its favorable effect did not explain the potential influence on the biomarkers in STZ-induced hyperglycemic rats, which was the objective of this investigation. On one side finding from our study will provide further prevention of diabetic complications via proposing a template for the development of novel drug-like candidates for or against biomarkers correlates with diabetic sequelae. In addition, it will also suggest the combining of CoQ10 with standard diabetes treatment, which may alleviate diabetic complications in another way.

## 2. Materials and experimental design

In caring for diabetic animals, we took into account the prominent characteristics of the disease, such as marked glycosuria, polydipsia, and polyuria. Therefore, significant attention is dedicated to providing access to drinking water and ensuring the comfort and hygiene of the animals' cages. Every effort has been made to alleviate suffering. The operations that potentially cause distress or pain were carried out in a separate room away from the other animals.

### 2.1. Experimental animals

Twenty-four male mature albino rats of Wister strain (250–350 g) were purchased from the animal house of the faculty of Science, Soran University. They were allowed to acclimate themselves to the new location for one week. The rats were provided with the standard rodent chow

(according to guidelines of the National Center for Drug Research and Quality Control, Baghdad) and water as often as necessary. The animals, housed three to four rats per cage, were situated within an air-conditioned room that met standard humidity conditions, light-dark cycle (12-hour light, 12-hour dark), and temperature (22˚C ± 2˚C). The rats were haphazardly allocated to one of three groups: normal control, diabetic control, or diabetic treatment group; each group consisted of eight rats.

## 2.2. Materials

Streptozotocin was purchased from Shanghai Macklin Biochemical Co., Ltd (Purity %97.5). CoQ10 200mg, which had been purchased from the pharmaceutical market, was manufactured by America (Nature's bounty Pharmaceutical Inc., USA.). Rat TNF-α ELISA Kit (48 tests, Reference Number: DZE201110765) was purchased from Sunred Biological Technology Co., Ltd., Shanghai, China. The rat resistin ELISA kit (Catalogue Number: SL0618Ra, 96 tests), rat omentin ELISA kit (Catalogue Number: SL0539Ra, 96 tests), and rat LPPLA2 ELISA kit (Catalogue Number: SL1066Ra, 96 tests) were purchased from Sunlong Biotech Co., LTD, China. Cardiac troponin I (cTnI) Kit, and Creatinine kinase MB (CK-MB) kit, were obtained from Hotgen Biotech company. The automatic hematology analyzer Swelab Alfa Standard (Boule Medical AB, Sweden) was used for hematological analysis.

## 2.3. Experimental design

After acclimatizing for 7 days, the rodents were randomly allocated into three groups:

Normal control (n = 7)

Diabetic control (n = 7)

Diabetic treatment group (treat with CoQ10, 5 gm per kg of food) (n = 7)

In the second and third groups (16 rats), we induced diabetes. Rats from Normal control (Group A) and Diabetic control (Group B) received the rat's standard diet without any supplementation. The CoQ10-enriched diet was given to the third group, which was the basal diet supplemented with 5 g of CoQ10 per kilogram of diet [44–48]. All groups had free access to their respective diets, and the treatments were continued for three weeks. The calculation of the sample size was based on the resource equation from earlier articles [49].

**2.3.1. Induction of diabetes.** The diabetic state induction in rodents depends on the injected dose of STZ, a diabetogenic agent that destroys pancreatic β-cells [50, 51]. The most often utilized approach is to inject a single STZ's dose (40–70 mg/kg) intraperitoneally into rats between the ages of 8 and 10 weeks [52]. The STZ-treated rat is a model for insulin-dependent, or type 1 diabetes mellitus (T1DM) [53]. We had carried out the procedure of diabetic induction according to the protocol which was written by Brian L. Furman [54]. The day before STZ injection, we removed food from feeders and fasted all rats for 6–8 hours prior to STZ administration. We dissolved STZ in a freshly prepared citrate buffer (0.1 M, pH 4.4–4.5) that was kept at a low temperature by putting it in an ice-cold vessel. Sixteen rats were intraperitoneally injected by STZ (45 mg/kg of body weight). The first rats' group (Group A) was intraperitoneally injected with an identical volume of citrate buffer solution (pH 4.4 to 4.5). Hypoglycemia can be anticipated to occur after STZ injection; therefore, they were supplying standard food and 10% sucrose water. Three of our animals died after STZ treatment due to hypoglycemia caused by extensive necrosis of pancreatic β-cells and an abrupt discharge of insulin, leading to lethal hypoglycemia, usually occurring within 48 hours of STZ administration [55]. Six rats remained in group B, and seven in group C. Another rat from the normal

group was injected with STZ, and on the first day of treatment, we had seven rats per group. On day two, after the STZ injection, the 10% sucrose drink was replaced with regular water.

Streptozotocin generates rat diabetic models within three days by damaging pancreatic β-cells [56]. After three days of diabetes induction, animals were fasted (6–8 hours (between the periods of 7 a.m. to 1–3 p.m.)), and their blood sugar levels were evaluated. Glucose levels were measured by Bayer Contour® TS Blood Glucose Monitoring System by testing blood from the tail of rats. Animals whose blood sugar levels exceeded 250 mg/dL were considered diabetic [57]. The manifestations of hyperglycemia were observed in rats likewise; polydipsia and polyphagia (the food and water consumption of the diabetic groups were dramatically noticeable more than normal control group), polyuria (the beds of diabetic groups had to be changed every day, but for normal rats every five days), and weight loss.

**2.3.2. Blood and serum collection.**   On the last day of the three-week treatment period, animals were starved overnight and sacrificed the subsequent day, and their blood samples were obtained. The procedure of serum collection began with the anesthetization of rats with a mixture of ketamine (a dose of 45 mg/kg) and xylazine (a dose of 5 mg/kg), then proceeded with a heart puncture using a sterile disposable syringe. A portion of blood was collected into tubes containing EDTA and immediately used to determine hematological parameters (white blood cell counts) and glucose levels. Another portion of blood was transferred into labeled gel tubes and left to coagulate at room temperature for 10–20 minutes. Then it was centrifuged between 2000 to 3000 rpm for 20 minutes and was applied to dislodge the clot, and the serum was separated for biochemical evaluation.

**2.3.3. Measurement of serum biomarkers.**   *A. Evaluation of TNF-α, resistin, omentin, and LPPLA2.* The interesting adipokines and some hyperglycemia-related markers were evaluated using rat enzyme linking immune-absorbent assay (ELISA) kits via microplate reader according to the absorbance principle. TNF-α, resistin, omentin, and LPPLA2 rat kits were purchased to assess TNF-α, resistin, omentin, and LPPLA2 according to the supplier's instructions and protocol of the respective kits.

Our ELISA kits for assessment of resistin, omentin, and LPPLA2 were designed to apply the Sandwich-ELISA technique. The micro-ELISA stripplates supplied in these kits have been pre-coated with an antibody specific to biomarkers (resistin, omentin, and LP-PLA2) standards or samples which are added to the wells of the appropriate micro-ELISA stripplate and mixed with the specific antibody. Then, a Horseradish Peroxidase (HRP)-conjugated antibody specific for each biomarker is injected into each well of the Microelisa stripplate, incubated, and free components are removed by washing. In each well, the TMB (3,3',5,5'-Tetramethyl-benzidine) substrate solution is added. TMB is a substrate for horseradish peroxidase (HRP), a typical antibody label used in ELISA and immunohistochemistry. Only those wells containing biomarkers and HRP conjugated tested biomarker (resistin, omentin, and LP-PLA2) antibodies will appear blue and then change yellow when the stop solution has been added.

Kit for assessment of TNF-α was designed to apply the double-antibody sandwich ELISA technique. Sample containing TNF-α is added to monoclonal antibody enzyme well, which is pre-coated with specific rat biomarker monoclonal antibody and incubated. Then biomarker antibodies labeled with biotin are added and combined with Streptavidin-HRP to form an immune complex. After that, another incubation is carried out and is washed again to remove the uncombined enzyme. After adding chromogen, solutions A and B, the color of the liquid will change to blue. Finally, because of the impact of acid, the color of wells is altered to yellow.

Measurement of the optical density (OD) is spectrophotometrically accomplished at a wavelength of 450 nm. According to the standards' concentration and the corresponding OD values, calculate the standard curve linear regression equation and then apply the OD values of the sample to the regression equation to calculate the corresponding sample's concentration.

*B. Evaluation of serum levels of cTnI, and CK-MB.* UP—Converting Phosphor Immunoassay Analyzer was used for quantitative measurement of serum concentration of cardiac troponin I (cTnI) and creatinine kinase MB (CK-MB) by using commercially available kits (Hotgen Biotech Co., Ltd). Serum or plasma Levels of these biomarkers are measured by a technique which is the combination of Up-converting Phosphor Technology (UPT) employing immunochromatography. Compared to other approaches, the technical data of UPT are uncomplicated to comprehend, there is no background interference with a specified signal, and the luminescence is persistent without a quench. Hotgen produced a POCT platform relying on UPT (UPT-3A) Series Immunoassay Analyzer and support reagents used in our study [58, 59].

The kit employs a combination of Up-converting Phosphor Technology (UPT) with sandwich immunochromatography. The reaction zone (T band) on the nitrocellulose membrane of the test cassette is coated with cTnI antibody, while the control zone (C band) is coated with goat anti-mouse antibody. When a diluted sample containing cTnI is added to the sample cavity in the test cassette, the capillary effect causes the fluid to flow to the other end. During the migration, cTnI in the sample first binds to the anti-cTnI antibody coated on the Up-converting Phosphor (UCP) nanoparticles, then bind to the cTnI antibody immobilized on the T line of the test cassette, forming antibody-antigen-antibody-UCP complex, and the rest UCP nanoparticles flow forward and bind to goat anti-mouse antibody on the C line, forming a secondary antibody-antibody-UCP complex. UCP particles emit visible light when excited by an infrared source. The intensity of the emission from the UCP particles at the T line and C line are measured simultaneously, and the ratio (T/C) of the emission intensity is proportional to the cTnI concentration in the sample. cTnI concentration is automatically calculated by reference to a calibration curve stored in the UCT system and displayed on the screen of the instrument. The concentrations of creatine kinase MB (CK-MB) were determined in the same approach as cTnI, which was mentioned above.

*C. Evaluation of hematological parameters.* Blood drawn through cardiac puncture was mixed by Blood Rolling Mixer machine and analyzed for hematological parameters such as lymphocyte percentage (LYM%), monocytes, eosinophils and basophils percentage (MID%), and granulocytes percentage (GRA%), using the automatic hematology analyzer Swelab Alfa Standard (Boule Medical AB, Sweden) at Ala private laboratory in Soran city.

## 2.4. Statistical analysis

Statistical analysis was performed using GraphPad Prism 9 (San CA, USA). Prior to conducting the analysis, we performed Normality and Lognormality tests to ensure that our data passed normality tests (alpha = 0.05) based on the Shapiro-Wilk tests. In order to compare the means of each parameter for the three groups (normal control = A, diabetic control = B, and CoQ10 treatment = C), we conducted One-Way ANOVA. In relation to multiple comparisons, the mean of columns A and C has been compared with columns B. Dunnett is recommended when making multiple comparisons using statistical hypothesis testing. All values were presented as mean ± standard error of mean, while the results at $P < 0.05$ were considered as significant.

## 3. Results

### 3.1. Effect of CoQ10 supplementation on blood glucose

As presented in Table 1 and Fig 1, the CoQ10 treatment group (5 g of CoQ10 per kilogram of diet) showed a significant reduction in blood glucose (P = 0.0012) compared to the diabetic animals without treatment. However, the concentrations of blood glucose in the CoQ10 treatment group were noticeably more than normal control group.

**Table 1. The impact of CoQ10 on blood glucose and adipokines in diabetic rats.**

| Parameters | Negative control (A) | Positive control (B) | CoQ10-treatmemt (C) | P-value | P-value |
|---|---|---|---|---|---|
| | | | | B vs. A | B vs. C |
| Glucose (mg/dl) | 95.90±9.057 | 468.5± 35.20 | 269.8 ± 38.50 | <0.0001 | 0.0012 |
| TNF-α (pg/ml) | 39.05±7.985 | 123.5 ±4.327 | 69.33 ±7.093 | <0.0001 | <0.0001 |
| Resistin (ng/ml) | 15.76±2.46 | 23.07±0.29 | 17.86 ±0.59 | 0.0026 | 0.0258 |
| Omentin (pg/ml) | 70.62±3.35 | 107.1 ±8.77 | 61.57±7.979 | 0.0108 | 0.0011 |

Values are expressed as mean ±SEM, P<0.05 indicates significant difference, NS indicates not significant

## 3.2. Effect of CoQ10 supplementation on serum levels of TNF-α, resistin, omentin, and LPPLA2

The results were evaluated at the end of the third week of the study in all groups, and the parameter values are presented in Tables 1 and 2. Also, Figs 1 and 2 show the changes in serum levels of TNF-α, resistin, omentin, and LPPLA2 in all groups of our experimental animals.

There was a significant difference between serum TNF-α levels of group A rats with group B and C. Compared with diabetic rats in group B, the serum level of TNF-α was dramatically lowered in the CoQ10-treated group (P<0.0001). Group A (Normal control) has the lowest value of serum level of TNF-α compared to both B and C groups, hyperglycemia cause up-regulation of TNF-α concentration in diabetic rats.

CoQ10 administration reduced serum resistin levels in group C rats (treatment group) which were significantly lower than in group diabetic control rats (p = 0.0258). In diabetic rats, we see increased resistin levels compared to the normal control group (P = 0.0026). Serum omentin level in the diabetic control group was significantly higher than both of the other two groups. The values of serum omentin in group C were significantly lower than in group B

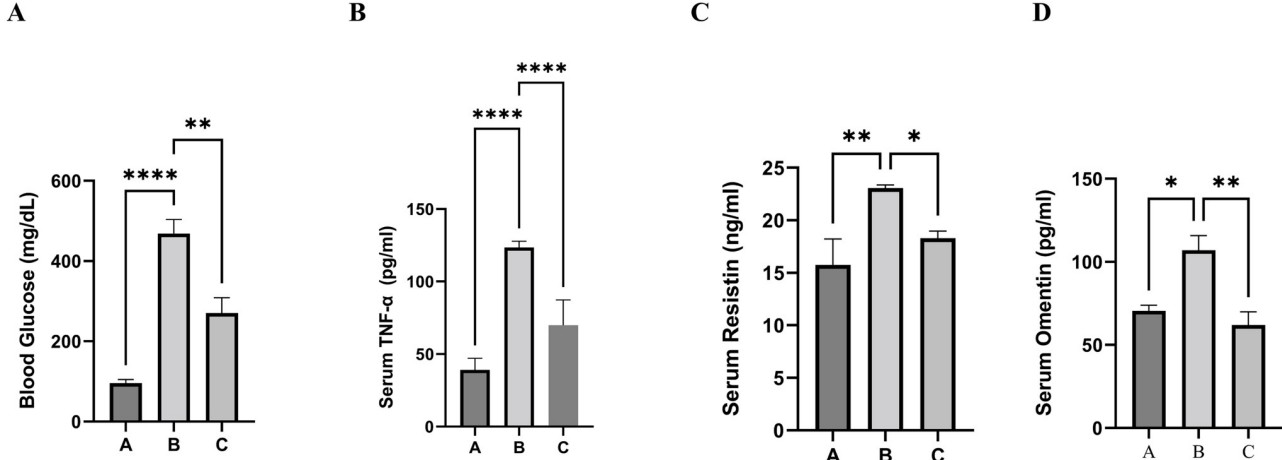

**Fig 1. Effect of CoQ10 on blood glucose levels and serum levels of resistin, TNF-α, and omentin.** After three weeks, blood glucose levels and serum levels of TNF-α, resistin, omentin, and LPPLA2 were measured in three groups of experimental rats; the Normal control group (A), the Diabetic control group (B), and the Diabetic group treated with CoQ10 (C). Glucose levels were measured by Bayer Contour® TS Blood Glucose Monitoring System. Serum levels of omentin, resistin, and TNF-α were evaluated by enzyme-linked immunosorbent assay. Data are presented as the mean ± SE (n = 7). Within the same duration of treatment, bars with different asterisks (∗, ∗∗) differ from each other significantly (P < 0.05). Non-asterisk bars do not differ significantly from one another (P > 0.05) within the same duration of treatment.

**Table 2. The effects of CoQ10 on LPPLA2 and cardiomyopathy markers in diabetic rats.**

| Parameters | Negative control (A) | Positive control (B) | CoQ10-treatmemt (C) | P-value | P-value |
|---|---|---|---|---|---|
| | | | | B vs. A | B vs. C |
| LPPLA2 (pg/ml) | 609.1±16.19 | 734.1±5.41 | 656.0±9.32 | 0.0005 | 0.0001 |
| Troponin (ng/ml) | 1.363±0.096 | 3.157±0.86 | 1.387±0.21 | 0.0460 | 0.0399 |
| CK-MB (ng/ml) | 22.14±7.40 | 54.75±12.00 | 21.86±4.64 | 0.0353 | 0.0229 |

Values are expressed as mean ±SEM, P<0.05 indicates significant difference, NS indicates not significant

(P = 0.0011). Group A (Normal control) has the lowest value of serum level of omentin; high blood sugar increased the level of omentin in hyperglycemic rodents.

There was a significant difference between serum LPPLA2 levels of group A rats with group B (P = 0.0005), and rats in the diabetic control group had the highest values. The serum levels of LPPLA2 in the treatment group were significantly lower than in group B (P = 0.0001).

### 3.3. Effect of CoQ10 supplementation on serum levels of cTnI, and CK-MB

As presented in Fig 2 and documented parameters values in Table 2, the highest serum values of cTnI and CK-MB levels were seen in diabetic control rats (Group B), indicated that high blood sugar caused an increment of their levels and CoQ10-treated downregulated their serum concentration. Serum cTnI levels were significantly decreased in the diabetic CoQ10-treated group (Group C) compared with group B (P = 0.0399). Also, CoQ10 lowered CK-MB levels in group C in comparison to diabetic control rats (P = 0.0229).

### 3.4. Effects of CoQ10 supplementation on hematological parameters

The hematological profile, as depicted in Table 3 and Fig 3, reveals a statistically significant difference (p < 0.05) between the diabetic control group (Group B) and the normal control group (Group A). Lymphocytes showed a significant decrease (P = 0.0125) in group diabetic control when compared with the normal control group throughout the experimental period. However, the dose of 5 mg of CoQ10 per kg of food did not significantly raise the level of lymphocytes from 57.30±4.330% in diabetic control rats to 60.14±3.240% in diabetic treated rats.

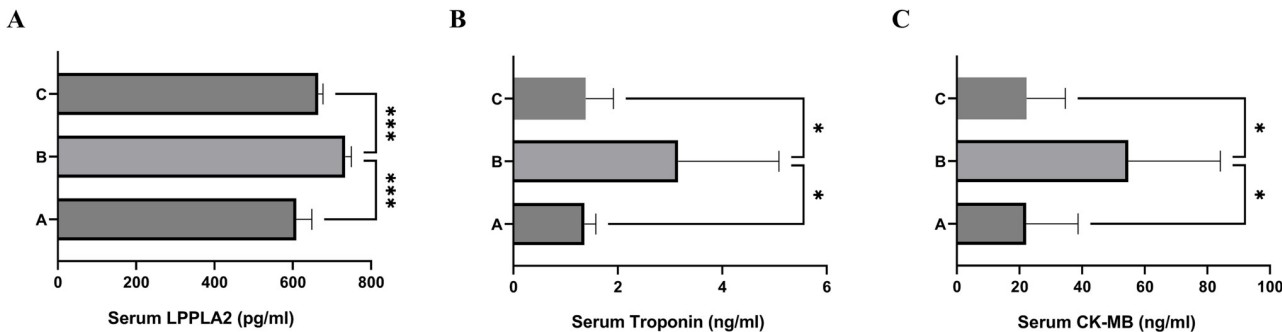

**Fig 2. Effect of CoQ10 on serum levels of LPPLA2, troponin, and creatinine kinase-MB.** After three weeks, serum levels of LPPLA2, cardiac troponin, and Creatinine kinase-MB were measured in three groups of experimental rats; the Normal control group (A) and the Diabetic control group (B), and the Diabetic group treated with CoQ10 (C). Serum levels of LPPLA2 were measured by enzyme-linked immunosorbent assay. Serum levels of cardiac troponin and creatinine kinase-MB were measured by UP—Converting Phosphor Immunoassay Analyzer. Data are presented as the mean ± SE (n = 7). Within the same duration of treatment, bars with different asterisks (∗, ∗∗) differ from each other significantly (P < 0.05). Non-asterisk bars do not differ significantly from one another (P > 0.05) within the same duration of treatment.

**Table 3. The influence of CoQ10 on the hematological profile of diabetic rats.**

| Parameters | Negative control (A) | Positive control (B) | CoQ10-treatmemt (C) | P-value | P-value |
|---|---|---|---|---|---|
| | | | | B vs. A | B vs. C |
| Lymphocytes% | 77.50±5.759 | 57.30±4.330 | 60.14±3.240 | 0.0125 | NS |
| Granulocytes% | 22.14±7.405 | 54.75±12.00 | 21.86±4.642 | 0.0071 | NS |
| Mid% | 22.14±7.405 | 54.75±12.00 | 21.86±4.642 | 0.0389 | NS |

Values are expressed as mean ±SEM, P<0.05 indicates significant difference, NS indicates not significant

Other hematological indices were also not significantly altered (Table 3 and Fig 3) between groups B and C. The parameter values show that CoQ10 decreased the percentage of mono-cytes, eosinophils, and basophils (MID%) in the treated group compared to the diabetic con-trol group, but that was nonsignificant. The same results were seen between groups A, B, and C related to the granulocyte levels, and their levels in group C were lower than in group B, and group A had the lowest levels of granulocytes.

## 4. Discussion

### 4.1. Effect of CoQ10 on blood glucose concentration

Our observations revealed that the administration of STZ led to a significant increase in blood glucose levels among the experimental rats, surpassing the normal range (Fig 1 and Table 1), thereby confirming their diabetic status. Streptozotocin is an antibiotic that destroys pancre-atic β-cells (Fig 4), and it is widely used experimentally to produce a model of T1DM. The results of β-cells destruction are insufficient insulin, hyperglycemia, polydipsia, and polyuria, which are all signs of human T1DM [54, 60].

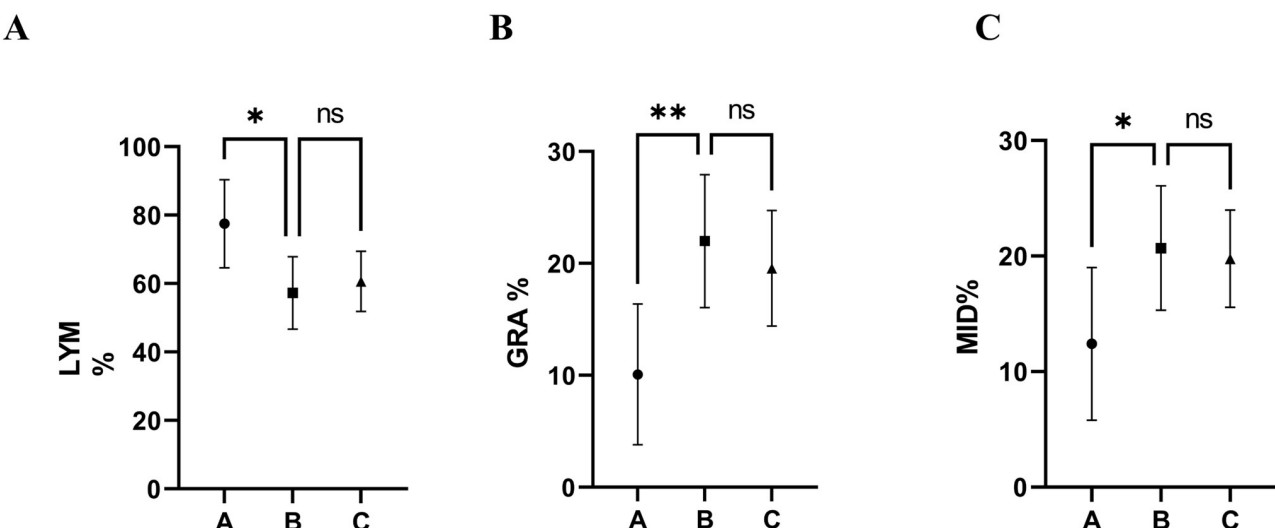

**Fig 3. Effect of CoQ10 on blood levels of hematological parameters, lymphocyte percentage (LYM%), granulocytes percentage (GRA%), and medium size cells percentage of monocytes, eosinophils and basophils (MID%).** After three weeks, blood levels of LYM%, MID%, and GRA% were measured in three groups of experimental rats, the Normal control group (A), the Diabetic control group (B), and the Diabetic group treated with CoQ10 (C). hematological parameters were evaluated by the automatic hematology analyzer Swelab Alfa Standard (Boule Medical AB, Sweden). Data are presented as the mean ± SE (n = 7). Within the same duration of treatment, bars with different asterisks (∗, ∗∗) differ from each other significantly (P < 0.05)—ns: not significant.

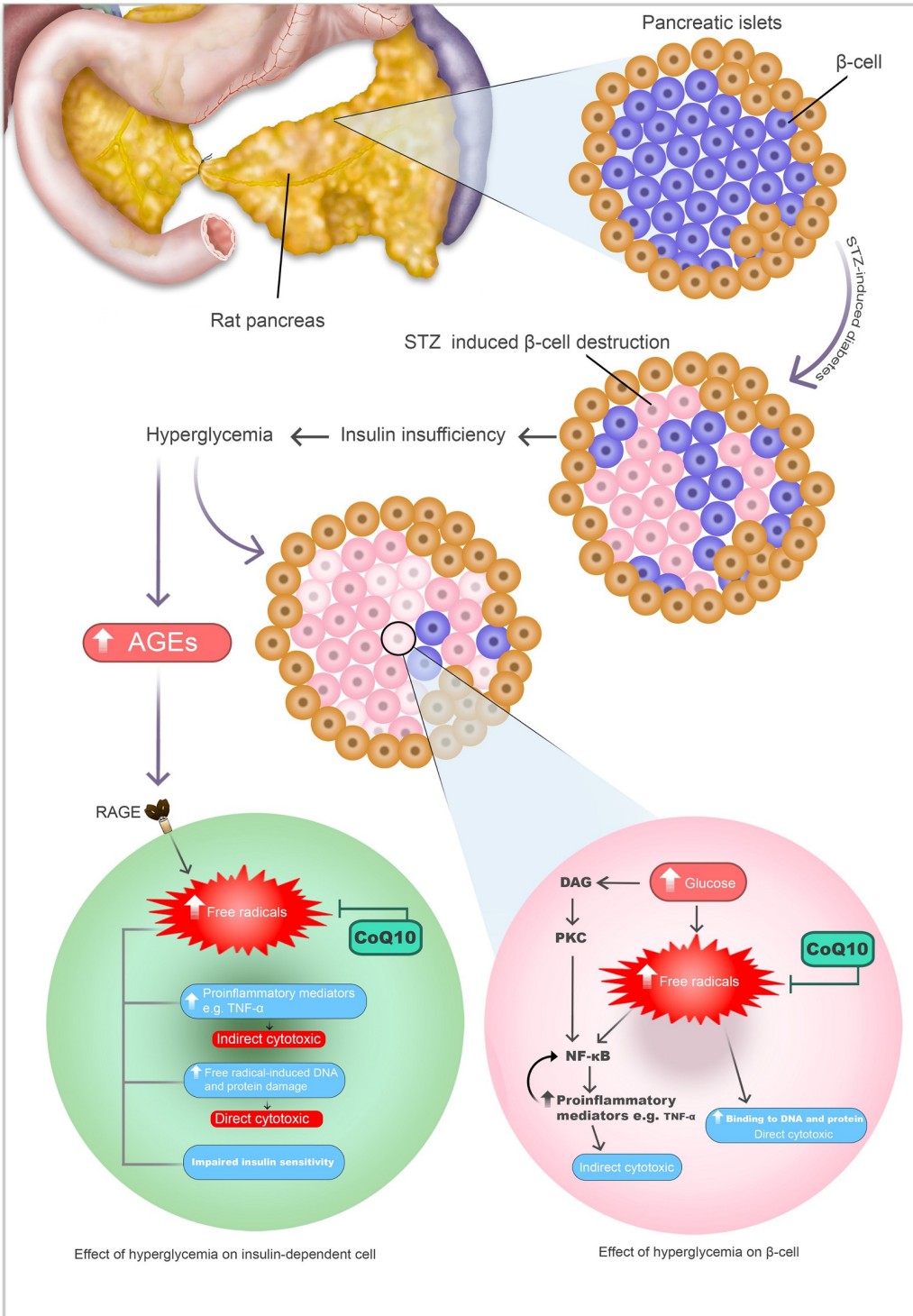

**Fig 4. Schematic diagram representing the mechanism of β-cell destruction by STZ and glucotoxicity, and the proposed mechanism of regeneration of β-cell function by CoQ10.**

Numerous studies have shown that the development of microvascular and cardiovascular diabetes problems is strongly influenced by oxidative stress. Pro-oxidative processes, which are frequently enhanced in hyperglycemia, have been characterized as several cascades of molecular mechanisms in several metabolic pathways [21, 61]. Oxidative stress had previously been identified as a substantial factor in the pathogenesis and complications of hyperglycemia [62].

Increased glucose levels lead to oxidative stress by increasing mitochondrial reactive oxygen species (Fig 4), nonenzymatic glycation of proteins, and glucose autoxidation [63]. The ramifications of persistent hyperglycemia, notably oxidative stress, have been extensively documented across various research studies. If left uncontrolled, it can potentially compromise the activity of antioxidant enzymes and other crucial antioxidant molecules [64].

Hyperglycemic status increases free radical production and impairs the endogenous antioxidant defense system [65] which resulting in production of AGEs. AGEs increase IL-6 secretion through ROS and NF-κB activation [66]. Also, ROS increase the activation of NF-κB in the cytoplasm and translocation of the signal transduction pathway into the nucleus. In inflammation and apoptosis, NF-κB acts as a key regulator. Phosphorylation of NF-κB induces pro-inflammatory cytokines such as TNF-α [67, 68]. As shown in Fig 4, this pathway, NF-κB activation by high glucose level and AGES, and the pro-inflammatory effect of TNF-α terminate in indirect cytotoxicity. Excess free radicals can cause harm to cellular lipids, proteins, or DNA, thereby disrupting their normal function and causing direct cytotoxicity [64, 69].

In the initial phase of STZ toxicity, some cells survive; hyperglycemic toxicity adds further impairment to these surviving cells' function; the mechanism of mitochondrial glucotoxicity in those cells that survive can thus be elucidated. As a result of glucose overload, a number of metabolic pathways are activated that not only attempt to dispose of excess glucose but also produce more reactive oxygen species, which leads to oxidative stress and cell death (Fig 4). It is suggested that all these pathways result in reactive oxygen species production, in conjunction with a low antioxidant capacity in cells, which are responsible for secondary diabetic β-cells failure [70]. In the presence of high levels of free radicals, pancreatic islet cells die and become dysfunctional (ROS-induced apoptosis).

The data in Fig 1 and Table 1 shows a lower blood sugar level in the CoQ10-treated group compared to the diabetic control group. As illustrated in schematic diagram Fig 4, it would be elucidated that the protective mechanism of CoQ10 is due to its ability in the neutralization of free radicals. Since NF-κB can be activated by ROS, the ability of CoQ10 to scavenge these free radical species may make an important contribution to its ability to inhibit the activation of NF-κB and could protect pancreatic β-cells [71]. As a result of this hypothesis, we focused on the antioxidative effect of CoQ10 on oxidative injury in pancreatic β-cells. It is likely that CoQ10 neutralizes free radicals and prevents their cytotoxic effect on surviving β-cells after STZ injection. As a result, Langerhans islets undergo quantitative and qualitative enhancements, thereby increasing their productivity. Consequently, this heightened productivity promotes augmented insulin production, ultimately contributing to a reduction in blood glucose levels. In diabetes, oxidative stress seems to be predominantly triggered by an elevation in the formation of reactive species coupled with a substantial depletion in antioxidant defenses [72, 73]. Fluctuations in redox balances, auto-oxidation of glucose, impaired antioxidant defense enzyme activities (such as catalase and superoxide dismutase), and decreased tissue concentrations of low molecular weight antioxidants (such as diminished vitamin E and glutathione) are potential causes of oxidative stress [74, 75]. Superoxide dismutase catalyzes the dismutation of two molecules of superoxide anion into hydrogen peroxide and molecular oxygen, rendering the potentially hazardous superoxide anion less harmful. Catalase facilitates the breakdown or reduction of hydrogen peroxide to water and molecular oxygen, effectively completing the detoxification step initiated by superoxide dismutase [76]. It is considered that elevated ROS

levels and a deterioration in antioxidant defense system activity significantly contribute to the development of diabetes complications [77, 78].

ROS and RNS cause oxidative damage to biological macromolecules (such as DNA, lipids, and proteins) and inhibit their normal function [79, 80]. Changes in membrane permeability, integrity, and flexibility, as well as adverse consequences and functional modification of membrane-bound proteins, can result from lipid peroxidation. Damage to the structural stability of proteins, resulting in the loss of catalytic properties of numerous enzymes and paralysis in regulating metabolic pathways, is another crucial aspect of oxidative stress [81]. Overproduction of RNS (nitrosative stress) can result in nitrosylation reactions which can affect the structure of proteins and impair their function [80, 82]. ROS combines with deoxyribose and nitrogenous bases in DNA, resulting in substantial oxidative reactions. These reactions can cause mutations, malignancy, necrosis, apoptosis, and hereditary diseases [81, 83].

The antioxidant CoQ10 helps reduce oxidative stress and protects the mitochondria in pancreatic β-cells [71]. Other researchers have demonstrated the impact, as mentioned above, of CoQ10 prescription [84, 85].

## 4.2. Effect of CoQ10 on serum TNF-α level

According to the data of our study, the increment of tumor necrosis factor-alpha (TNF-α) has been confirmed in diabetic rats. Previous studies also have reported serum up-regulation of TNF-α levels in human and experimental animal models of diabetes [86–90].

Fig 4 illustrates how high glucose levels and AGEs activate NF-κB via free radicals and the diacylglycerol-PKC signal transduction pathway, which induces the production of TNF-α, a key risk factor for diabetic complications and insulin resistance. The resulting TNF-α further activates NF-κB. NF-κB stimulates redox-sensitive signaling pathways leading to inflammation and fibrosis through indirect damage [91, 92].

TNF-α triggers cell adhesion via activating endothelial cells, overexpression of Major histocompatibility complex (MHC) class I and II within the islet, T cells activation antigen-presenting cells (APCs), and homing leukocyte [93–96], which altogether cause endothelial destruction (Fig 5).

A study found that cardiomyocytes cultured under high glucose levels expressed more NF-κB and TNF-α. In this study, researchers demonstrated that high glucose levels could substantially influence the structure and function of cultured cardiomyocytes, resulting in cardiac hypertrophy via the PKC signal transduction pathway, which could lead to diabetic cardiomyopathy. In diabetes, NF- κB plays a key role in the pathogenesis of vascular complications [97–99]. The mechanism and pathway of glucotoxicity's detrimental impact will likely be proposed from our and previous findings, as shown in Figs 4–6.

In addition to the NF-κB -TNF-α pathway, ROS can directly oxidize and damage DNA, proteins, and lipids, which is believed to contribute directly to diabetic complications and cytotoxicity [63, 100], which showed in Figs 4–6 as the direct cytotoxic outcome.

An investigation showed that resistin secretion induces activation and upregulation of TNF-α [101]. Human resistin is proven to promote the synthesis and release of TNF-α [102].

Researchers found that resistin activates macrophages to produce TNF-α (Fig 5). The mentioned effects of resistin, along with the finding that resistin activated the NF-κB transcription factor, supports the notion that macrophage signal cascades contribute to obesity and diabetes [102].

Our study has confirmed the results of other researchers [89, 90, 103] that administration of CoQ10 in rats led to reduced serum TNF-α levels, which may have been attributed to CoQ10's declining effects on resistin levels which are generally heightened in patients with

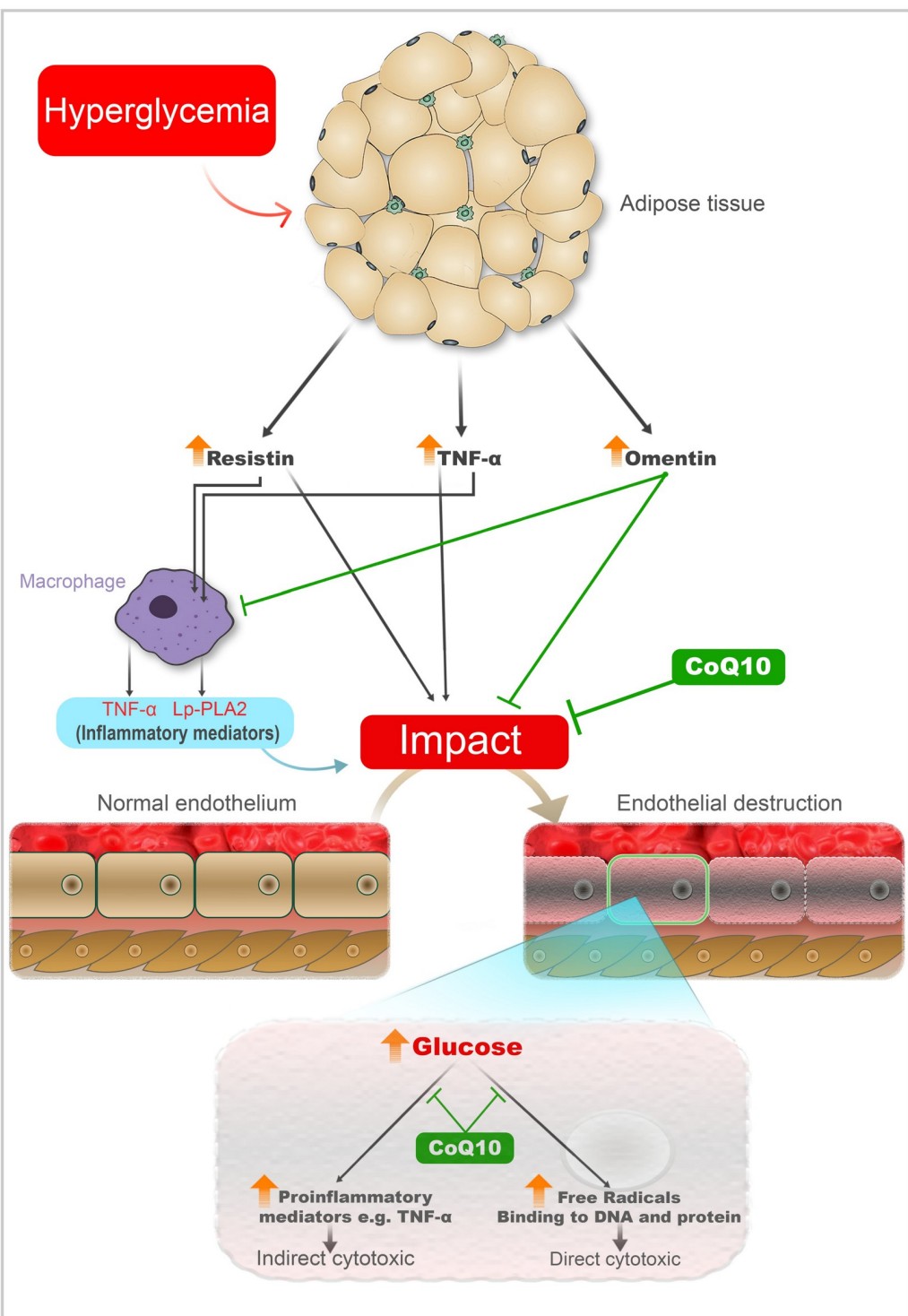

**Fig 5. A schematic diagram depicting the proposed pathway wherein hyperglycemia triggers the release of adipokines that influence endothelial inflammation.** TNF-α and resistin cause inflammation and damage of endothelium, while omentin protects endothelium of blood vessels, LPPLA2 in response to the adipokines (TNF-α and resistin) causes further damage of endothelium.

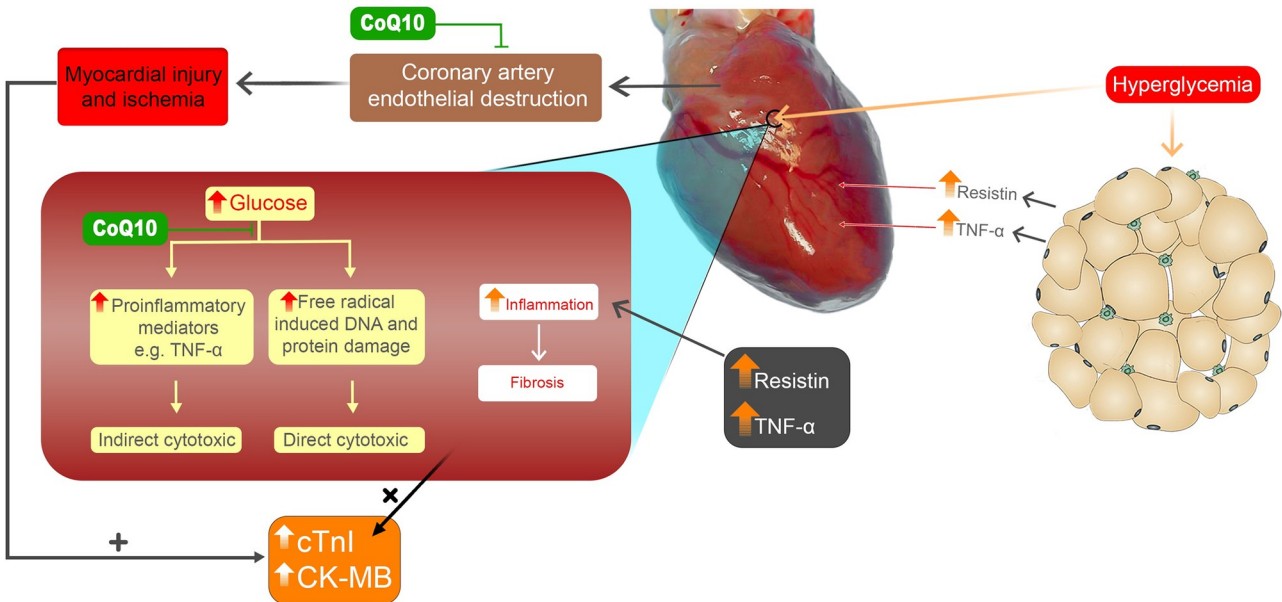

**Fig 6. Schematic diagram illustrates our findings concerning the consequences of hyperglycemia on heart muscles.**

diabetes. Other researchers found that antioxidants block the ability of various stimuli to trigger NF-κB activation [104]. The inhibition effect of CoQ10, as a free radical (or ROS) scavenger in our study, is illustrated in Figs 4–6, the possible mechanisms of the reduction of TNF-α levels following the administration of CoQ10 is summarized. CoQ10 would counteract the effect of free radicals in β-cells, insulin-dependent cells, endothelial cells, and cardiac cells. Through its ability to neutralize and scavenge free radicals, CoQ10 decreases stimulation of NF-κB and resistin in the body, as well as TNF-α production.

### 4.3. Effect of CoQ10 on serum resistin level

In diabetic rats, resistin serum levels showed a significant increase, which is correlated positively with TNF-α serum levels. The level of the former is reduced by a substantial amount in the group that received CoQ10 (Table 1 and Fig 1). Other studies found that diabetic subjects have higher fast blood sugar and resistin levels than normal control rats [105, 106]. However, a study found that resistin level decreases in mice with hyper-insulinemic states. As a result of acute hyperglycemia, resistin is up-regulated in various adipose depots (as presented in Fig 5). In our study, the STZ-induced diabetic control rats were hyperglycemic and hypo-insulinemic, so they had a high resistin level, which parallels the findings of other studies [105–108].

The biomarker resistin is likely to be an indicator of a unique relationship between metabolic signals, atherosclerosis, and inflammation that is associated with human atherosclerosis [109, 110]. In acute coronary syndrome and ischemic heart disease, high resistin levels could be a biomarker of injury [111]. The novel adipokine resistin is suspected of contributing to vascular endothelial dysfunction (Fig 5) through its pathological effects on endothelium, such as upregulation of vascular cell adhesion molecule 1 (VCAM-1), endothelin 1 (ET-1), and induction of superoxides and eNOS activation [112].

Prior studies have identified Toll-Like Receptor 4 (TLR4) and Adenylyl Cyclase-Associated Protein 1 (CAP1) as the membrane receptors for human resistin. These receptors are notably

present on macrophages and the vascular membrane of rodents. However, rodents also possess two other receptors, the Tyrosine Kinase-like Orphan Receptor 1 (ROR1) and the N-terminally truncated Decorin (DCN), which exist on the adipose cell membrane [113].

Some human investigations declared the role of resistin in the pathogeneses of inflammation and its contribution to atherogenesis impacts on vascular endothelium and smooth muscle cells [114, 115], we postulated that the stimulation impact of resistin on macrophages causes the secretion of TNF-α and LPPLA2, which are pro-inflammatory and inflammatory mediators and have negative consequences on vascular endothelial cells, as demonstrated in Fig 5.

Upon searching online recently, we found that the first human study assessing CoQ10's influence on resistin was performed in 2021 on nondiabetic subjects (human volunteers); supplementation of CoQ10 reduced serum resistin [116]. Using STZ-induced diabetic rats, we are the first to investigate the effect of CoQ10 on resistin. Interestingly, we found that rats treated with antioxidants display a lower resistin level than diabetic rats.

Similarly, resistin concentrations in the blood are inversely correlated with total antioxidant capacity, and antioxidant supplementation could dramatically lower resistin levels, according to Ozarda et al. and Bo et al. [117, 118], and we demonstrated in Fig 5, CoQ10 would prevent the production and release of resistin from adipose tissue and macrophages and inhibits NF-κB pathway. In addition, as demonstrated above, CoQ10 has a hypoglycemic effect and would increase insulin function through the recovery of β-cells function (Fig 4).

## 4.4. Effect of CoQ10 on serum omentin level

According to our study, serum omentin levels among diabetic rats increased in comparison with normal controls. Additionally, the CoQ10-treated group had a lower level of omentin than the diabetic rats without treatment. Similar to our findings which are shown in Fig 2, Nurten et al. also found that increased omentin values were associated with hyperglycemia [119]. Adipose tissue secretes omentin, which inhibits the formation of necrotic cores and the expression of pro-inflammatory cytokines within an atherosclerotic lesion [120, 121].

In STZ- induced diabetic rats, with a low insulin level, we observed a high level of omentin. In contrast, in the normal control group, with a normal insulin level, their serum omentin levels were less than diabetic rats. Also, other studies conducted on the human and rat model of type 2 diabetes revealed that omentin values for the diabetic group were significantly higher than those detected in the non-diabetic group [122, 123]. As seen in Fig 5, the impact of glucotoxicity on adipocytes could cause the release of omentin, which would be the consequence of insulin insufficiency.

We observed that serum omentin declined after CoQ10 was administered to the study subjects. It is more likely that CoQ10 indirectly lowers the blood omentin level by increasing insulin sensitivity and directly by decreasing blood sugar levels in diabetic rats. Besides, CoQ10 can protect β-cells from STZ toxicity, as has already been mentioned.

Some researchers mentioned the existence of omentin receptors on macrophages. Thus, As demonstrated in Fig 5, omentin can inhibit macrophages to release pro-inflammatory cytokines, TNF-α and resistin [121].

## 4.5. Effect of CoQ10 on serum LPPLA2 level

The serum lipoprotein-associated phospholipase A2 (LPPLA2) levels were significantly higher in the rats injected with STZ than in the normal control group (p<0.05). Several investigations reported elevated LPPLA2 levels when diabetic rats were compared with nondiabetic rats,

which agrees with our findings [124, 125]. First-ever analysis of CoQ10's effects on LPPLA2 in STZ-induced diabetic rats was conducted in our study.

Circulating LPPLA2 is considered an indicator of inflammation and is associated with vascular dysfunction [126]. Among patients with stable CAD, a high level of LPPLA2 is associated with endothelial dysfunction and arterial stiffness as an independent risk factor [127]. In the CoQ10 treated group, the level of LPPLA2 in serum was significantly lower than in the diabetic group using the Rat LPPLA2 ELISA kit. Hence, this study's findings suggest that CoQ10 may be beneficial to diabetics in addition to standard anti-diabetic treatments to reduce cardiovascular complications.

### 4.6. CoQ10 exhibits the ability to neutralize free radicals, consequently downregulating intracellular pathways responsible for LPPLA2 production by macrophages. Moreover, as previously elucidated, this treatment is expected to reduce the blood levels of resistin and TNF-α, thereby mitigating their stimulating effects on macrophages. Consequently, subjects treated with CoQ10 may exhibit decreased LPPLA2 concentrations, as depicted in the schematic diagram (Fig 5). Effect of CoQ10 on serum cardiac biomarkers

We measured biochemical indicators of myocardial injury; cardiac troponin I (cTnL) and creatine kinase-MB (CK-MB) levels in all groups of the experimental rats. We noticed the increment of all the cardiac markers in the diabetic control group and, the results revealed that CoQ10 prescription could decrease their levels in group C.

According to the diagram (Fig 6), glucotoxicity can stimulate adipocytes to release resistin and TNF-α, causing inflammation and, ultimately, apoptosis of cardiac tissue. In light of our study results, the reasons for the attenuation of diabetic cardiomyopathy by CoQ10 are as follows. Firstly, the supplement would prevent direct and indirect cytotoxicity. Secondly, it would protect pancreatic β-cells and have hypoglycemic properties, thereby lowering glucotoxicity's influence on raising blood adipokines; consequently, CoQ10 would protect vascular endothelial cells and minimize myocardial injury and ischemia as explained earlier in this article. Lastly, CoQ10 could reduce inflammation and apoptosis in heart tissue.

**4.6.a. Effect of CoQ10 on serum cTnI.** In our research highly specific biomarker for cardiac muscle, cardiac troponin I (cTnL), has been measured. There has been reported a positive link between myofibrillar breakdown and higher serum levels of cTnI and CK-MB [128]. The cardiac troponins T and I (cTnT and cTnI) have been shown to have higher specificity and sensitivity and are superior to creatine kinase-MB (CK-MB) as markers for diagnosing myocardial necrosis [129]. Troponins, particularly cTnI, are frequently employed in veterinary medicine and are extremely specific and sensitive indicators of heart abnormalities and lesions [130].

Our investigation revealed evidence of diabetes-related heart damage, characterized by elevated serum concentrations of cTnI and CK-MB in the diabetic control group. These findings are supported by previous studies indicating a correlation between elevated serum cTnI and CK-MB levels and experimentally induced cardiomyopathy in rats [131, 132]. We supposed that CoQ10 might have a cardioprotective effect by preventing hyperglycemia-induced cardiomyocyte damage. Interestingly, treatment with CoQ10 remarkably reduced the systemic levels of cTnI and CK-MB. Our data have shown the increment of cTnI levels in diabetic animals, and its levels in the CoQ10-treated rats were lower than in the diabetic subjects without treatment.

Supplementation with CoQ10 would defend the myocardium through its antioxidant activity, as was proven by improving different biochemical markers and oxidative status and restoring the myocardium's structural integrity and function [133]. Another study presented essential findings that orally administered CoQ10 could significantly increase CoQ10 levels in the human mitochondria and cardiac muscles. The therapeutic effect of increasing CoQ10 levels at such sites is enhanced protection of myofilaments and mitochondria from oxidative stress, resulting in efficient energy production and improved contractile recovery in vitro following hypoxia-reoxygenation stress [134].

**4.6.b. Effect of CoQ10 on serum CK-MB.**   In our study, we measured the CK-MB levels in all groups of experimental rats. CK-MB is a more sensitive and specific biomarker of myocardial damage, which rises in the presence of heart tissue injury [129, 135]. Similar to our findings, another study found diabetic rats had higher serum CK-MB levels than normal rats, indicating that inflammatory reaction, myocardial injury, and myocardial fibrosis were significantly exacerbated in diabetic rats [136]. In the current investigation, we noted that the serum level of CK-MB in the CoQ10 treated rats was lower than in the diabetic control rats. In agreement with our work, previous studies confirm our results that CoQ10 administration can decrease enzyme activities in plasma, demonstrating that CoQ10 treatment can decrease CK-MB activity in the experimental animals [137, 138]. Our research confirms that CoQ10 may protect against fibrosis and cardiac remodeling by lowering cardiac markers.

## 4.7. Effect of CoQ10 on hematological parameters

There are no studies pertaining to CoQ10's effects on hematological variables, but several studies have publicized other antioxidants' effects on the parameters. Concerning lymphocytes, we obtained similar results, which were recorded in the other studies that the lymphocytes from diabetic controls were significantly less than those from normal controls groups [139, 140]. But in contrast to our findings, in another work, the increase in the levels of lymphocyte count in diabetic rats was documented, and they declared that it could be a result of the damaging effects of STZ [141]. In line with our findings, a study corroborated that the administration of antioxidants led to an increased percentage of lymphocytes, eosinophils, and neutrophils in the antioxidant-treated group compared to the untreated group [142].

Another research revealed similar results to our experiment, which found that the level of neutrophil and eosinophil increases in hyperglycemic rats compared to normal rats. However, in discrepancy to our data, they documented that there was no effect of their antioxidant prescription on lymphocyte, neutrophil, or eosinophil count in all treated groups, but in our animal model, CoQ10 improved the percentages of them [143]. The raise in MID percentage could be due to the release of adipokines, such as TNF- and resistin, which can activate leukocytes, as illustrated in Fig 5.

Our study, meanwhile, has some limitations. Due to the deadline for the master project and limited financial capacity and cost of kits, we were unable to obtain NF-kB kits in accordance with the timetables. Shortage of histopathology materials and facility at the Hawler Medical University hindered us from doing histological examinations of the pancreas and heart to validate our biochemical test results. Because of budgetary constraints, we were unable to send tissue samples from rats to laboratories in other countries.

## 5. Conclusions

Hyperglycemic rats had increased serum levels of adipokines (omentin, resistin, and TNF-α) and cardiomyopathy markers (cardiac troponin I and Creatine kinase-MB). In relation to hematological markers, GRA% and MID increased, while LYM% declined. CoQ10

administration improved both the biomarkers mentioned above and hyperglycemia Hematological parameters were altered by CoQ10 supplementation, but the change was not statistically significant. Through modulation of endogenous antioxidant defenses, CoQ10 showed potential benefits in streptozotocin-induced diabetic rats by preventing cardiovascular complications. Moreover, when combined with conventional diabetes treatments, CoQ10 may contribute to ameliorating the consequences of diabetes-related complications.

## Acknowledgments

The authors acknowledge Hawler Medical University for providing one of the authors the opportunity to study as a master's candidate in the pharmacology department and conduct research. The authors acknowledge Soran University (in particular Sarbast A. Mahmud) for providing the necessary facilities required for performing a part of the study in their animal house.

## Author Contributions

**Formal analysis:** Yousif Jameel Jbrael, Badraldin Kareem Hamad.

**Funding acquisition:** Yousif Jameel Jbrael.

**Investigation:** Yousif Jameel Jbrael.

**Methodology:** Yousif Jameel Jbrael, Badraldin Kareem Hamad.

**Project administration:** Badraldin Kareem Hamad.

**Visualization:** Yousif Jameel Jbrael.

**Writing – original draft:** Yousif Jameel Jbrael.

**Writing – review & editing:** Yousif Jameel Jbrael, Badraldin Kareem Hamad.

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
