## [Decision Letter · Decision Letter 0]

4 Jan 2023

PONE-D-22-20964Ameliorating impact of coenzyme Q10 on the profile of adipokines, cardiomyopathy, and hematological markers correlated with the glucotoxicity sequelae in diabetic ratsPLOS ONE

Dear Dr Yousif Jameel Jbrael, 

Thank you for submitting your manuscript to PLOS ONE. After careful consideration, we feel that it has merit but does not fully meet PLOS ONE’s publication criteria as it currently stands. Therefore, we invite you to submit a revised version of the manuscript that addresses the points raised during the review process.

We look forward to receiving your revised manuscript.

Kind regards,

Naguib Bin Salleh, MBBS PhD

Academic Editor

PLOS ONE

Journal Requirements:

1 Please ensure that your manuscript meets PLOS ONE's style requirements, including those for file naming. The PLOS ONE style templates can be found at

2. As part of your revision, please complete and submit a copy of the Full ARRIVE 2.0 Guidelines checklist, a document that aims to improve experimental reporting and reproducibility of animal studies for purposes of post-publication data analysis and reproducibility: https://arriveguidelines.org/sites/arrive/files/documents/Author%20Checklist%20-%20Full.pdf (PDF). Please include your completed checklist as a Supporting Information file. Note that if your paper is accepted for publication, this checklist will be published as part of your article.

Additional Editor Comments (if provided):

Reviewer 1

Dear author

Thanks a lot for your submission

In abstract:

1- In material and method part please more explain about your work.

2- There is some grammatical error need to revise.

In introduction:

1- Please more explain the side effects of diabetes

2- In last paragraph, please explain about importance of your research.

In material and method:

1- Based on what evidence you choose the sample size?

2- More explain the biochemical assay analyses

In results:

1- Please present the all data of your study (table2 and tanle3)

2- Please more explain about the all data of the tables

In discussion

1- Please more discuss about the oxidative stress in diabetes

2- Please more discuss about the effect of stress on diabetic patient

3- Please explain about the limitation of your study.

Reviewer 2

Comments

1. Line 49…. The exact pathophysiological mechanisms remain unknown. This sentence is incorrect since there are abundant information with regards to pathophysiological mechanisms underlying DM and its complications.

2. Line 59-61… objectives should be placed at the end of the introduction.

3. Other reports indicating the benefits of Coenzyme Q10 on health should be included.

4. Line 86-94. There appears to be three objectives of the study with identifying the role of coenzyme Q10 being one of the objectives.

5. Authors should clearly state the gap of knowledge that exist pertaining to the role of coenzyme Q10 in ameliorating the effect of glucotoxicity and then provide a clear hypothesis to fulfill this knowledge gap. This is then followed by the study objectives.

6. Besides coenzyme Q10, are there any other supplements like coenzyme Q10 which is used to overcome diabetic complications?

7. Please provide animal ethics number.

8. Line 114. Should be the animals were housed in standardized housing condition

9. There should be a positive control group i.e., diabetic rats treated with glibenclamide (standard oral anti-diabetic agent) in order to compare the efficacy of coenzyme Q10.

10. This study model is a type-1 induced diabetic model as only STZ was given. This type of model should be highlighted in the title. In addition, some information with regards to type-1 DM should be given in the introduction section including its complications and the difference between type-1 and type 2 DM

11. What is the purpose of giving sucrose drink after STZ injection? This should be mentioned in the text.

12. Authors conducted the investigation only by analyzing serum levels of various markers. The results are not adequate as they are not supported by few other evidence such as histopathology ex. Histopathology of the pancreas, heart.

13. The rats body weight, food and water intakes need to be measured too. This is important as we would expect that in type 1 DM, there will be loss of body weight in contrast to type 2 DM which is usually associated with obesity.

14. Serum levels of insulin and HbAIc should also be measured.

15. Levels of NFkB in tissues should also be measured.

16. There are lots of speculations/postulations involved in the discussion which should be supported by real evidences. The findings are mostly preliminary. Authors should provide more evidences to support their argument such as histological evidences especially the pancreas and the heart, tissue inflammatory marker levels (TNFa and NF-kb), insulin levels etc.

Reviewers' comments:

Reviewer's Responses to Questions

**Comments to the Author**

1. Is the manuscript technically sound, and do the data support the conclusions?

Reviewer #1: Yes

2. Has the statistical analysis been performed appropriately and rigorously? 

Reviewer #1: Yes

3. Have the authors made all data underlying the findings in their manuscript fully available?

Reviewer #1: Yes

4. Is the manuscript presented in an intelligible fashion and written in standard English?

Reviewer #1: Yes

5. Review Comments to the Author

Reviewer #1: Dear author

Thanks a lot for your submission

In abstract:

1- In material and method part please more explain about your work.

2- There is some grammatical error need to revise.

In introduction:

1- Please more explain the side effects of diabetes

2- In last paragraph, please explain about importance of your research.

In material and method:

1- Based on what evidence you choose the sample size?

2- More explain the biochemical assay analyses

In results:

1- Please present the all data of your study (table2 and tanle3)

2- Please more explain about the all data of the tables

In discussion

1- Please more discuss about the oxidative stress in diabetes

2- Please more discuss about the effect of stress on diabetic patient

3- Please explain about the limitation of your study.

6. PLOS authors have the option to publish the peer review history of their article (what does this mean?). If published, this will include your full peer review and any attached files.

Reviewer #1: No

---

## [Author Response · Author response to Decision Letter 0]

1 Apr 2023

Rebuttal letter

Response to reviewers

We are appreciative to the editor and reviewers for evaluating our manuscript. According to the reviewers’ comments, we have made major revisions to the manuscript, and we believe that it has significantly improved. 

Hereby we give a point-by-point response to the comment/question, please, follow our responses:

Reviewer 1

In abstract:

1-In material and method part please more explain about your work.

Answer: We agree with the reviewer’s comment and have altered the content in the materials and methods section of the abstract accordingly to better describe our work. However, in order to adhere to the 300-word limit, we had to make a small modification.

2-There is some grammatical error need to revise.

Answer: Grammatical errors were revised.

In introduction:

1-Please more explain the side effects of diabetes

Answer: We agree with the reviewer's comment and now acknowledge that the introduction should have contained more details about diabetes adverse effects. We provided further information regarding diabetic complications.

2- In last paragraph, please explain about importance of your research. 

Answer: In the submitted paper, we had in some way addressed the significance of our research; however, we agree with the reviewer and have revised the text accordingly to emphasize the importance of our study.

In material and method:

1-Based on what evidence you choose the sample size? 

Answer: We choose the sample size based on a literature review. 

Charan, J. and Biswas, T., 2013. How to calculate sample size for different study designs in medical research?. Indian journal of psychological medicine, 35(2), pp.121-126.

In animal studies, the power analysis method for calculating sample size is suggested. When it is not possible to establish the effect size and the standard deviation, an alternative to the power analysis method is the ‘resource equation’ approach, which determines the permissible range of the error degrees of freedom (DF) in an analysis of variance (ANOVA). In this method a value E is calculated based on decided sample size. The value if E should lie within 10 to 20 for optimum sample size. If a value of E is less than 10 then more animal should be included and if it is more than 20 then sample size should be decreased.

E = Total number of animals – Total number of groups

2- More explain the biochemical assay analyses

Answer: We thank you for advising to suggest more explain the biochemical assay analyses. We elaborate on the biochemical assay analysis and provide further information regarding our laboratory procedures.

In results:

1-Please present the all data of your study (table2 and tanle3)

Answer: The linear regression equation for the calibrations curve is calculated by using concentration and optical density (OD) values of the standards, and then apply the OD values of the sample to the regression equation to get the sample's concentration. Hematological machine (Automatic hematology analyzer Swelab Alfa Standard) automatically generate parameters percentages and we included them in table 3. We attached the file of our data.

The mean value of each group’s biomarkers was determined and compared to the mean values of other groups. In order to compare the means of each parameter for the three groups, we conducted One-Way ANOVA. In relation to multiple comparisons, the mean of columns A and C has been compared with columns B.

2- Please more explain about the all data of the tables 

Answer: We are grateful for your comment. We revise it.

In discussion:

1-Please more discuss about the oxidative stress in diabetes

Answer: Based on this comment, we determined it would be beneficial to discuss the role of oxidative stress in diabetes in further detail.

2- Please more discuss about the effect of stress on diabetic patient

Answer: We appreciate your recommendation. We have now included the impacts of oxidative stress on diabetes patients in the Discussion section of the manuscript.

3- Please explain about the limitation of your study.

Answer: We thank the reviewer's critical but valuable comment. We mentioned the limitations of our research.

Reviewer 2

1. Line 49…. The exact pathophysiological mechanisms remain unknown. This sentence is incorrect since there are abundant information with regards to pathophysiological mechanisms underlying DM and its complications.

Answer: The precise pathophysiological mechanisms behind diabetes mellitus and its consequences were uncertain based on reference of our literature search.

Example:

Yaribeygi, H.; Sathyapalan, T.; Atkin, S. L.. Sahebkar, A. Molecular mechanisms linking oxidative stress and diabetes mellitus. Oxidative medicine and cellular longevity. 2020;2020

However, we revised the text, and we added detailed information in regard pathophysiology mechanism having been known so far.

2. Line 59-61… objectives should be placed at the end of the introduction.

Answer: Thank you; we made revisions.

3. Other reports indicating the benefits of Coenzyme Q10 on health should be included.

Answer: We do appreciate your advice on the revision, so thank you. We mentioned other studies indicating the beneficial properties of Coenzyme Q10 for health.

4. Line 86-94. There appears to be three objectives of the study with identifying the role of coenzyme Q10 being one of the objectives.

Answer: The current study investigates the relationship between glucotoxicity state and the level of some endogenous ligands, particularly adipokine biomarkers, vascular inflammation markers (such as Lipoprotein-associated phospholipase A2 (LPPLA2)). We also examined the correlations between hematological variables, cardiomyopathy biomarkers, and the state of glucotoxicity. Additionally, we evaluated at how the levels of the aforementioned markers are impacted by CoQ10, and we made the hypothesis that CoQ10 would ameliorate their profiles while also attenuating the risk of diabetes complications.

5. Authors should clearly state the gap of knowledge that exist pertaining to the role of coenzyme Q10 in ameliorating the effect of glucotoxicity and then provide a clear hypothesis to fulfill this knowledge gap. This is then followed by the study objectives.

Answer: 

Gap: There is a general consensus that oxidative stress is one of the serious contributors to the progression of diabetes complication. In diabetic rats, excessive antioxidant defense system consumption and lipid peroxidation have been identified, and in recent years, considerable emphasis has been devoted to the central and critical involvement role of oxidative stress in the etiology of numerous diabetic complications. No study or very little studies are available to evaluate the correlation of glucotoxicity with our study biomarkers (some endogenous ligands, particularly adipokine biomarkers, vascular inflammation marker (Lipoprotein-associated phospholipase A2 [LPPLA2]), hematological variables, and cardiomyopathy biomarkers). There is no research linking CoQ10's antioxidant activity to endogenous ligands or biomarkers which having role in the mechanism of diabetes complications.

Clear hypothesis: We propose that there is a correlation between glucotoxicity and levels of our examined markers (some endogenous ligands, particularly adipokine biomarkers, vascular inflammation markers (such as Lipoprotein-associated phospholipase A2 [LPPLA2]), hematological parameters, and cardiomyopathy biomarkers). Also, we hypothesize that there is a correlation between antioxidant effect of CoQ10 with the levels of above-mentioned ligands or biomarkers in favor of attenuating diabetic complications.

6. Besides coenzyme Q10, are there any other supplements like coenzyme Q10 which is used to overcome diabetic complications?

Answer: We appreciate the reviewer’s thoughtful idea. We include details about other supplements like coenzyme Q10, which is used to alleviate diabetes complications.

7. Please provide animal ethics number

Answer: Ethical approval for this study was obtained from Hawler Medical University’s institutional ethics committee (protocol numbers, HMU-EC-PH 150921-455).

8. Line 114. Should be the animals were housed in standardized housing condition

Answer: According to our literature search, the standard room temperature range for rat housing between 20 - 26° C is recommended. Other conditions were also standardized for housing the rats. 

9. There should be a positive control group i.e., diabetic rats treated with glibenclamide (standard oral anti-diabetic agent) in order to compare the efficacy of coenzyme Q10.

Answer: Due to the fact that there were a very limited number of rats (particularly male rats) at the time of research at our animal home, so that we only had 24 rats to separate into three groups; normal control group, diabetic control group, and CoQ10-treatment group. We compare the CoQ10-treatment group to normal control, diabetic control because we aimed to evaluate the efficacy and potency of CoQ10. Therefore, we were not able to compare CoQ10-treatment group with other treatment group (standard oral anti-diabetic agent like glibenclamide).

Furthermore, because there is such a wide range of anti-diabetic medicines (as known first drug of choice is metformin), so the numbers of groups had to be expanded. Because of limitation of rats’ number, especially male rats, we could not add other positive treatment groups for anti-diabetic drugs. In the future, we had the plans to carry research in our university animal house. We will conduct studies comparing CoQ10-treatment group with other anti-diabetic drugs groups (particularly new hyperglycemic agents) or in a combination with them.

10. This study model is a type-1 induced diabetic model as only STZ was given. This type of model should be highlighted in the title. In addition, some information with regards to type-1 DM should be given in the introduction section including it complications and the difference between type-1 and type 2 DM.

Answer: We studied the impact of glucotoxicity in hyperglycemic state in type-1 induced diabetic rat model. Regarding to our search in literature, many other researchers have done research in the STZ-induced type one animal model, but they did not include it in the title. We revised the manuscript and added information about diabetes complications.

11. What is the purpose of giving sucrose drink after STZ injection? This should be mentioned in the text.

Answer: Sorry, but the reason for providing a sucrose drink after a STZ injection has previously been stated, kind reviewer.

“Hypoglycemia can be anticipated to occur after STZ injection; therefore, they were supplying standard food and 10% sucrose water. Three of our animals died after STZ treatment due to hypoglycemia caused by extensive necrosis of pancreatic β‐cells and an abrupt discharge of insulin, leading to lethal hypoglycemia, usually occurring within 48 hours of STZ administration.”

12. Authors conducted the investigation only by analyzing serum levels of various markers. The results are not adequate as they are not supported by few other evidence such as histopathology ex. Histopathology of the pancreas, heart.

Answer: In response to another reviewer’s comment, it has been noticed that we had limitation to do histopathology examinations. Lack of histopathology materials and facility at the Hawler Medical University hindered us from doing histological examinations of the pancreas and heart to validate our biochemical test results.

13. The rats body weight, food and water intakes need to be measured too. This is important as we would expect that in type 1 DM, there will be loss of body weight in contrast to type 2 DM which is usually associated with obesity.

Answer: Frankly, we did not measure precisely the amount of food and water consumed by the rats but as long as we monitored the amount of food and drinking according of body weight of rats, because the objectives of our study were the evaluation of the hyperglycemia effect on biomarkers, ligands, and parameters which we mentioned above as well as an assessment of the changes in their levels after prescription of CoQ10. We observed the manifestation of hyperglycemia in rats likewise; polydipsia and polyphagia (the food and water consumption of the diabetic groups were dramatically noticeable more than normal control group), polyuria (we had to change bed of diabetic groups every day, but for normal rats every five days), and weight loss (We did not consider it necessary to include the weight measurement table in the manuscript. The weight measurement chart is provided below. It can be seen the considerably reduction of average weight in diabetic rats).

(Normal control = A, diabetic control = B, and CoQ10 treatment = C)

14. Serum levels of insulin and HbAIc should also be measured.

insulin

We didn't repeat study of the CoQ10 effect on insulin levels because that had previously been conducted, and We have included the references to two articles below:

Modi K, Santani DD, Goyal RK, Bhatt PA. Effect of coenzyme Q10 on catalase activity and other antioxidant parameters in streptozotocin-induced diabetic rats. Biological trace element research. 2006 Jan;109:25-33.

Maheshwari, R.A., Parmar, G.R., Hinsu, D., Seth, A.K. and Balaraman, R., 2020. Novel therapeutic intervention of coenzyme Q10 and its combination with pioglitazone on the mRNA expression level of adipocytokines in diabetic rats. Life Sciences, 258, p.118155.

We did not assess HbA1c in the current study due to the 21-day research duration, regarding that HbA1c has usually served as an indicator of glycemic control over the preceding 2-to-3-month period.

15. Levels of NF-кB in tissues should also be measured.

Answer: Due to the deadline of the master project and limited financial capacity and cost of the kits, we did not measure NF-kB . In addition, there are numerous studies having already focused on this biomarker, hence, we were not interested to evaluate the same biomarker, instead, we did measure TNF alpha (they have mutual up-regulation effect ) which is used as background and template biomarker and or endogenous ligand in our study .

16. There are lots of speculations/postulations involved in the discussion which should be supported by real evidences. The findings are mostly preliminary. Authors should provide more evidences to support their argument such as histological evidences especially the pancreas and the heart, tissue inflammatory marker levels (TNFa and NF-кB), insulin levels etc. 

Answer: Due to reasons mentioned above and as previously indicated, we were unable to do histological evaluations of the pancreas and heart at Hawler Medical University because of a shortage of histopathology materials and facility, there has already been addressed as a limitation of our research. Study of novel biomarkers and adipokines was the main priority for us, therefore we invest our resources (tuition fee) in their study.

Several researchers have investigated NF-кB, therefore we did not repeat their work. As it is known, the transcription factor NF-кB stimulates the transcription of proinflammatory cytokines including TNF-α, and also NF-кB can be activated by TNF-α. Because TNF- α and NF-кB have a mutual impact, we assessed TNF- α instead of NF-кB as a template (and a reference) for our investigation.

With kindest regards

---

## [Decision Letter · Decision Letter 1]

3 Nov 2023

PONE-D-22-20964R1Ameliorating impact of coenzyme Q10 on the profile of adipokines, cardiomyopathy, and hematological markers correlated with the glucotoxicity sequelae in diabetic ratsPLOS ONE

Dear Dr. Jbrael,

Thank you for submitting your manuscript to PLOS ONE. After careful consideration, we feel that it has merit but does not fully meet PLOS ONE’s publication criteria as it currently stands. Therefore, we invite you to submit a revised version of the manuscript that addresses the points raised during the review process.

We look forward to receiving your revised manuscript.

Kind regards,

Mohammed Fouad El Basuini, Professor

Academic Editor

PLOS ONE

Journal Requirements:

Reviewers' comments:

Reviewer's Responses to Questions

**Comments to the Author**

1. If the authors have adequately addressed your comments raised in a previous round of review and you feel that this manuscript is now acceptable for publication, you may indicate that here to bypass the “Comments to the Author” section, enter your conflict of interest statement in the “Confidential to Editor” section, and submit your "Accept" recommendation.

Reviewer #2: All comments have been addressed

Reviewer #3: All comments have been addressed

2. Is the manuscript technically sound, and do the data support the conclusions?

Reviewer #2: Yes

Reviewer #3: Yes

3. Has the statistical analysis been performed appropriately and rigorously? 

Reviewer #2: Yes

Reviewer #3: Yes

4. Have the authors made all data underlying the findings in their manuscript fully available?

Reviewer #2: Yes

Reviewer #3: Yes

5. Is the manuscript presented in an intelligible fashion and written in standard English?

Reviewer #2: No

Reviewer #3: Yes

6. Review Comments to the Author

Reviewer #2: Thank you for sending the revised manuscript. However, there are some points that require attention and corrections:

1. In the abstract's background section, it would be helpful to provide a brief explanation of glucotoxicity and its relevance to diabetes mellitus. This will provide more context for readers who may not be familiar with the term.

2. In the abstract's conclusion section, it would be beneficial to summarize the main findings of the study concisely. Additionally, discussing the potential implications of the results and highlighting any limitations or future directions for research would be useful.

3. Ensure that the references cited in the manuscript follow a consistent format. Maintain uniformity in the citation style throughout the text.

4. In the methodology, please include specific details about the conditions of acclimatization and measures taken to ensure the animals' well-being during this period. This should encompass information such as the number of animals in each cage, environmental enrichment, and health and behavior monitoring.

5. Please provide more information regarding the ELISA procedure. Include details such as dilutions, incubation times, washing steps, and other relevant protocol-specific information for each ELISA kit used in the study.

6. The manuscript would benefit from proofreading to correct minor grammatical and typographical errors, improving clarity and readability.

Reviewer #3: The paper is well-designed and prepared. All comments have been addressed, and there are no further comments. Accept in the present form.

7. PLOS authors have the option to publish the peer review history of their article (what does this mean?). If published, this will include your full peer review and any attached files.

Reviewer #2: **Yes: **Giribabu Nelli

Reviewer #3: No

---

## [Author Response · Author response to Decision Letter 1]

8 Dec 2023

Rebuttal letter

Response to reviewers

We express our gratitude to the editor and reviewers for their evaluation of our manuscript. Based on the feedback provided by the reviewers, we have implemented revisions to the manuscript, resulting in what we believe is a considerable improvement.

We will provide a detailed response to each comment or question, addressing them point by point. Please follow our responses accordingly:

Reviewer 2

We wish to express our sincere gratitude for the valuable feedback you have shared. Your insights are immensely appreciated and will undoubtedly contribute to the improvement of our manuscript.

1-In the abstract's background section, it would be helpful to provide a brief explanation of glucotoxicity and its relevance to diabetes mellitus. This will provide more context for readers who may not be familiar with the term.

Answer: We concur with the reviewer's comment and have made changes to the content in the background section of the abstract to succinctly elucidate glucotoxicity and its relevance to diabetes mellitus. Nevertheless, to comply with the 300-word limit, we made a minor adjustment.

2. In the abstract's conclusion section, it would be beneficial to summarize the main findings of the study concisely. Additionally, discussing the potential implications of the results and highlighting any limitations or future directions for research would be useful.

Answer: We discussed the beneficial effects of the antioxidant CoQ10 that we investigated, particularly its role in improving diabetic complications. Due to the 300-word limit, we were unable to provide further explanation within our abstract. We addressed the potential implications of our study results in the final paragraph of the introduction section and in the conclusion of our article.

3. Ensure that the references cited in the manuscript follow a consistent format. Maintain uniformity in the citation style throughout the text.

Answer: Thank you for your valuable feedback regarding the consistency of citation formatting in our manuscript. We have taken your comments seriously and have made concerted efforts to ensure uniformity in the referencing style throughout the document.

To address this concern, I meticulously reviewed all references cited in the manuscript and utilized the reference management tool, EndNote, to enforce adherence to the PLOS ONE style (Vancouver). This involved a comprehensive check of each citation within the text, cross-referencing them with the bibliography or reference list, and ensuring that they align with the specified guidelines of the Vancouver style.

4. In the methodology, please include specific details about the conditions of acclimatization and measures taken to ensure the animals' well-being during this period. This should encompass information such as the number of animals in each cage, environmental enrichment, and health and behavior monitoring.

Answer: We appreciate your guidance on the revisions; we made changes accordingly. We specified the number of animals in each cage and also included details about specific details about the conditions of acclimatization, environmental enrichment, as well as health and behavior monitoring in Lines 166-184.

5. Please provide more information regarding the ELISA procedure. Include details such as dilutions, incubation times, washing steps, and other relevant protocol-specific information for each ELISA kit used in the study.

Answer: Thank you for your query regarding the ELISA procedure. We appreciate your interest in the details and protocol specifics.

In our manuscript, we have provided a concise overview of the ELISA procedure in lines 251-268. Additionally, we have attached the PDF file containing the Rat LP-PLA2 ELISA Kit instruction to this email for your reference. If you believe that including specific details from the kit catalogues in the study manuscript would enhance its comprehensiveness, we are more than willing to incorporate this information in our next revision. Your input is highly valued, and we aim to address all essential details to strengthen the manuscript.

Furthermore, an extensive search was conducted within the PLOS ONE journal to gather relevant information. Specifically, we tailored our procedure to align with the methodology outlined in previously published articles; the DOI of one such article is provided below:

https://doi.org/10.1371/journal.pone.0202797

6. The manuscript would benefit from proofreading to correct minor grammatical and typographical errors, improving clarity and readability.

Answer: Thank you for your constructive feedback. We acknowledge the need for proofreading to rectify any minor grammatical and typographical errors in the manuscript. Ensuring clarity and readability is pivotal, and we are committed to enhancing these aspects.

In our forthcoming revision, we will diligently conduct a thorough proofreading process to address these issues and refine the overall quality of the manuscript.

Reviewer 3

The paper is well-designed and prepared. All comments have been addressed, and there are no further comments. Accept in the present form.

Thank you for your positive feedback and acceptance recommendation. We appreciate your time and effort in reviewing our article. Your insights were invaluable.

Best regards,

---

## [Editor Report · Decision Letter 2]

19 Dec 2023

Ameliorating impact of coenzyme Q10 on the profile of adipokines, cardiomyopathy, and hematological markers correlated with the glucotoxicity sequelae in diabetic rats

PONE-D-22-20964R2

Dear Dr. Jbrael,

We’re pleased to inform you that your manuscript has been judged scientifically suitable for publication and will be formally accepted for publication once it meets all outstanding technical requirements.

Kind regards,

Mohammed Fouad El Basuini, Professor

Academic Editor

PLOS ONE
---

## [Editor Report · Acceptance letter]

4 Jan 2024

PONE-D-22-20964R2 

PLOS ONE

Dear Dr. Jbrael, 

I'm pleased to inform you that your manuscript has been deemed suitable for publication in PLOS ONE. Congratulations! Your manuscript is now being handed over to our production team.

Kind regards, 

on behalf of

Dr Mohammed Fouad El Basuini 

Academic Editor

PLOS ONE